# BEYOND SIZE: HOW GRADIENTS SHAPE PRUNING DECISIONS IN LARGE LANGUAGE MODELS

## ABSTRACT

Large Language Models (LLMs) with a billion or more parameters are prime targets for network pruning, which aims to reduce a portion of the network weights without compromising performance. Prior approaches such as Weights Magnitude, SparseGPT, and Wanda, either concentrated solely on weights or integrated weights with activations for sparsity. However, they overlooked the informative gradients derived from pretrained large language models. In this paper, we present a novel sparsity-centric pruning method for pretrained LLMs, termed **G**radient-**b**ased **L**anguage **M**odel **P**runer (GBLM-Pruner). GBLM-Pruner leverages the first-order term of the Taylor expansion, operating in a training-free manner by harnessing properly normalized gradients from a few calibration samples to determine the importance pruning score, and substantially outperforms competitive counterparts like SparseGPT and Wanda in multiple benchmarks. Intriguing, after incorporating gradients, the unstructured pruning method tends to reveal some structural patterns post-pruning, which mirrors the geometric interdependence inherent in the LLMs' parameter structure. Additionally, GBLM-Pruner functions without any subsequent retraining or weight updates to maintain its simplicity as other counterparts. Extensive evaluations on LLaMA-1 and LLaMA-2 across various language benchmarks and perplexity show that GBLM-Pruner surpasses magnitude pruning, Wanda (*weights+activations*) and SparseGPT (*weights+activations+weight update*) by significant margins. We further extend the proposed approach to the ViT model to demonstrate its broad applicability. Our code and models will be publicly available.

## 1 INTRODUCTION

Large Language Models (LLMs) like OpenAI's GPT series (Radford et al., 2018; 2019; Brown et al., 2020a; OpenAI, 2023), BERT (Devlin et al., 2018), LLaMA-1/2 (Touvron et al., 2023a;b) and others have made significant strides in recent years, leading to a paradigm shift in various domains of artificial intelligence and especially in natural language processing (OpenAI, 2023; Anil et al., 2023; Touvron et al., 2023b) and multimodal learning (Alayrac et al., 2022; Li et al., 2023). Many industries have integrated LLMs into their workflow, such as in chatbots (OpenAI, 2023), content generation (Anil et al., 2023), code completion tools (e.g., GitHub Copilot) (Chen et al., 2021), gaming narratives (Todd et al., 2023), and assistive technologies (Zdravkova et al., 2022), etc. While enjoying the powerful and capable of impressive generalization, LLMs come with a set of challenges and disadvantages. The presence of an abundance of parameters, large memory consumption, and the resultant high computational cost during inference present several concerns in real-world applications. Previous literature proposed multiple solutions to address these disadvantages, such as model distillation (Hinton et al., 2015), quantization (Jacob et al., 2018), model pruning (Han et al., 2016), hardware acceleration (Chen et al., 2020), etc.

Among them, pruning refers to the removal of certain weights or even whole neurons/layers from an LLM based on specified criteria, such as the smallest weights. A pruned model can maintain similar performance with fewer parameters, resulting in a reduction in storage and computational requirements. Inducing nonstructural sparsity in pruning is a widely embraced method aimed at minimizing the memory requirements of neural networks with only a minimal sacrifice in accuracy. Pruning methods stand out as notably simple and efficient mechanisms for model compression, serving to eliminate weights contingent on their significance. Reduced models are not only more

conveniently dispatched to edge devices like mobile phones but also exhibit substantially lower energy consumption, a sizable portion of energy is expended in transferring model parameters from a device's long-term storage to its memory (Dao et al., 2022).

However, given the constraints of training-free conditions, existing solutions for pruning LLMs primarily employ either weight magnitude pruning (Han et al., 2015a; 2016) or a combination of magnitude and activation pruning (Frantar & Alistarh, 2023; Sun et al., 2023). While these methods are substantiated with empirical ablations and experiments, they are, to a degree, either too complex to use like SparseGPT by computing matrix inverses and updating weights, or heuristic and lack profound theoretical depth and justification like Wanda regarding the efficacy, especially concerning the application to the recently developed, highly advanced large language models.

In this study, we tackle the aforementioned complexity and interpretability challenges in LLM pruning methods by presenting a simple yet effective approach named `GBLM-Pruner` (Gradient-Based Language Model Pruner) that can be well explained in theory using the adapted optimal brain surgeon (OBS) (Hassibi et al., 1993b). This method proficiently prunes LLMs to significant levels of sparsity, eliminating the necessity to alter the residual weights. Specifically, we employ normalization of gradients across various samples to formulate an indicator matrix. This matrix can serve as activations and can either replace or supplement them. This method maintains simplicity over SparseGPT (Frantar & Alistarh, 2023) while showcasing enhanced robustness and improved interpretation than Wanda (Sun et al., 2023) on large language models compared to both *magnitude* pruning and *magnitude + activation* pruning. Furthermore, it is notable that although we employ gradients in our approach, there is no necessity for retraining or any updates to parameters.

**Difference to Previous Gradient-based Methods.** Although the use of gradients has been studied in the context of pruning, earlier methods (Molchanov et al., 2016b; Sanh et al., 2020a) used gradients in the context of transfer learning to obtain a pruned model that preserves the accuracy of the downstream task. This work is the first attempt to study the use of gradients for one-shot pruning of language models with billions of parameter while maintaining the zero-shot generalization capabilities of the language models to diverse downstream tasks. Additionally our proposed method does not require weight update, which makes our proposed method computationally efficient and applicable for large language models with billions of parameters like LLaMA-1-30B and LLaMA-2-70B.

We conducted extensive empirical evaluations of `GBLM-Pruner` on LLaMA-1 and 2 (Touvron et al., 2023a;b), among the most influential families of open-sourced LLMs. The findings crossing various language benchmarks and perplexity from our investigation highlight that `GBLM-Pruner` is proficient in identifying effective sparse networks directly from pretrained LLMs, eliminating the need for retraining or weight updates. Notably, `GBLM-Pruner` substantially surpasses magnitude pruning and the recently introduced methods designed by *weights+activations* or *weights+activations+weight update*. Our contributions in this work form a foundational basis for ensuing advancements in this domain. Furthermore, we advocate for continued exploration aimed at unraveling the complexities of sparsity within LLMs through underexplored *gradients*, and highlighting that this is the first attempt to understand the importance of gradient information both theoretically and empirically, and introduce a simple gradient-based solution for LLMs pruning in a training-free manner. Finally, we further extend our approach to the other domain of Vision Transformer (Dosovitskiy et al., 2020) to demonstrate its effectiveness.

## 2 Approach

### 2.1 Prior Solutions

**Weights Magnitude.** Magnitude pruning, which retains weights of significant absolute values, is the predominant approach for weight pruning. This approach usually generates an unstructured sparsity and has been employed across various architectures spanning computer vision (Han et al., 2015a; 2016) and language processing (Gale et al., 2019b). Furthermore, it has recently become integral to the lottery ticket hypothesis (Frankle & Carbin, 2018).

**Weights and Activations.** SparseGPT (Frantar & Alistarh, 2023) conceptualizes the problem of pruning large language models by addressing a local, layer-wise reconstruction problem. The approach for determining pruning metrics and the process for updating weights in SparseGPT draws inspiration from the Optimal Brain Surgeon (OBS) (Hassibi et al., 1993b) approach. The pruning

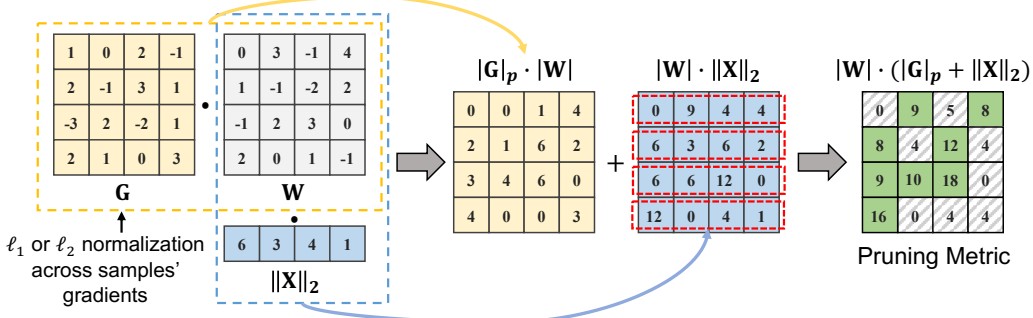

Figure 1: Illustration of the proposed method `GBLM-Pruner`. Given a weight matrix, $\mathbf{W}$, a gradient matrix, $\mathbf{G}$, and an input feature activation, $\mathbf{X}$, the computation of weight importance is executed through an elementwise multiplication of the magnitude of weight and the $\ell_1$ or $\ell_2$ norm of the gradients across multiple samples, denoted as $\|\mathbf{G}\|_p \cdot |\mathbf{W}|$, optionally, it is promotable to add the multiplication of weight and the $\ell_2$ norm of input activations, denoted as $|\mathbf{W}| \cdot \|\mathbf{X}\|_2$.

metric employed within SparseGPT is defined as follows:

$$\mathbf{W}_{\mathrm{m}}[i,j] = \frac{|\mathbf{W}[i,j]|^2}{\mathrm{diag}\left(\mathbf{H}^{-1}\right)[j,j]} \tag{1}$$

where $\mathbf{H} = \left(\mathbf{X}^T\mathbf{X} + \lambda\mathbf{I}\right)$ is the Hessian matrix, and $\mathbf{H}^{-1}$ is the inverse Hessian matrix. $\mathbf{W}_{\mathrm{m}}$ is the pruning importance metric for a given weight $\mathbf{W}$, and $[i,j]$ is the element at index $i,j$ of the matrix.

Wanda (Sun et al., 2023) suggests assessing the significance of each individual weight by calculating the product of its magnitude and the norm of the corresponding input feature. More precisely, the score for a given weight, $\mathbf{W}[i,j]$, is determined as follows:

$$\mathbf{W}_{\mathrm{m}}[i,j] = |\mathbf{W}[i,j]| \cdot \|\mathbf{X}[:,j]\|_2 \tag{2}$$

where the elementwise product between the weight magnitude and the norm of input activations is performed within each row in $\mathbf{W}$.

## 2.2 GRADIENTS MATTER

**Gradients.** According to Optimal Brain Damage (LeCun et al., 1989) and Optimal Brain Surgeon (Hassibi et al., 1993b), gradients and higher order derivatives are naturally correlated to the importance of weights for LLM pruning, which is the theoretical basis of our approach. However, they ignore the gradients in their pruning framework under the assumption that gradients of the fully trained network are small and do not provide any additional information when the higher-order terms are considered. Our work shows that gradients are still crucial and provide non-trivial information.

Previous gradient-based structured pruning methods, such as feature map pruning (Molchanov et al., 2016a), channel pruning (Yang et al., 2022), and head pruning (Michel et al., 2019) utilize the first-order Taylor approximation of the loss $\mathcal{L}$ around activation $z = 0$ or weight $w = 0$ as the importance score, the formulation is:

$$\mathbf{W}_{\mathrm{m}} = \mathbb{E}_{\boldsymbol{x}\sim\mathbf{X}}\left|\frac{\partial\mathcal{L}(\boldsymbol{x})}{\partial\mathbf{A}}\mathbf{A}\right| \tag{3}$$

where $\mathbf{X}$ is the sampled data distribution and $A$ is either activation matrix $Z$ or weight matrix $W$. Most of these structured pruning methods are proposed for transfer learning to a particular task and require significant finetuning on the specific task to maintain the model performance. In contrast, our work proposes how to leverage gradient information to do unstructured and N:M semi-structured pruning without any subsequent weight update. Additionally, we illustrate the integration of activations into our pruning metric through the use of a scaling factor for best performance. Furthermore, our pruned model is task-agnostic and generalizable to any downstream task as showcased by the Zero-shot evaluation on several tasks included in the Etheuther AI lm-evaluation harness benchmark (Gao et al., 2021).

**Pruning Metric.** As illustrated in Algorithm 1, consider a layer in LLMs characterized by the weight $\mathbf{W}$, possessing a shape of $(d_{out}, d_{in})$. In the context of Transformer models, this layer has the gradient $\mathbf{G}$, exhibiting the same shape of weight $\mathbf{W}$. We propose evaluating the importance of each individual weight by normalizing the corresponding gradients across different samples and then computing the product of its magnitude with the weights. More precisely, the importance score attributed to a specific weight, $\mathbf{W}[i, j]$, is determined as follows:

$$\mathbf{W}_{\mathrm{m}}[i,j] = |\mathbf{W}[i,j]| \cdot \|\mathbf{G}[:,i,j]\|_p \quad (4)$$

While competitive results can be achieved with gradients solely, we can combine feature activations to get better performance, which form our final pruning metric as shown in Equation 5:

$$\mathbf{W}_{\mathrm{m}}[i,j] = |\mathbf{W}[i,j]| \cdot \left( \alpha \cdot \|\mathbf{G}[:,i,j]\|_p + \|\mathbf{X}[:,j]\|_2 \right)$$
$$(5)$$

where $\alpha$ is the scaling factor used to account for the small magnitude of gradients, which makes the contribution of gradient balanced to the large magnitude of activations.

---

**Algorithm 1** The `GBLM-Pruner` algorithm

---

$\mathbf{W} \leftarrow$ weight matrix $\in (d_{out}, d_{in})$
$\mathbf{X} \leftarrow$ activation matrix $\in (N \times L, d_{in})$
$\mathbf{G} \leftarrow$ gradient matrix $\in (N, d_{out}, d_{in})$
$p \leftarrow$ sparsity ratio $\in (0, 1)$
$\mathbf{W}_{\mathrm{m}} \leftarrow$ pruning metric $\in (d_{out}, d_{in})$
$\mathbf{M} \leftarrow$ pruning mask $\in (d_{out}, d_{in})$

**for** $i \in (1, d_{out})$ **do**
    **for** $j \in (1, d_{in})$ **do**
        $\mathbf{W}_{\mathrm{m}}[i,j] = (|\mathbf{W}[i,j]| \cdot \|\mathbf{G}[:,i,j]\|_p + |\mathbf{W}[i,j]| \cdot \|\mathbf{X}[:,j]\|_2)$
    **end for**
**end for**

**for** $i \in (1, d_{out})$ **do**
    $\mathbf{M}[i,:]$ = mask of $p\%$ weights with smallest $\mathbf{W}_{\mathrm{m}}[i,:]$
**end for**

$\mathbf{W}[\mathbf{M}] = 0$

---

**Pruning Granularity.** The granularity of pruning is pivotal in unstructured pruning, owing to the fact that varying granularities yield disparate pruning patterns. Previously, unstructured magnitude pruning approaches have leveraged both layer-wise and global pruning. In these methods, weights are contrasted either within the same layer or throughout the entirety of the model. Through a comprehensive study, we observe that the highest accuracy is achieved when weights are analyzed on a column-wise basis. This is because each column serves as a constituent component in output activation. This insight is consistent with the findings presented in Sun et al. (2023).

## 2.3 A THEORETICAL ANALYSIS

In this section, we have revisited and refined the Optimal Brain Surgeon (OBS) framework (Hassibi et al., 1993b) framework by incorporating considerations of the gradient, i.e., the first-order term in Taylor approximation. The closed-form solution of the increase in error for removing a weight from the model, given by this analysis serves as the fundamental basis for our novel gradient-based pruning metric. For the sake of simplicity, we will consider weights and gradients as one-dimensional vectors denoted by $\boldsymbol{w}$ and $\boldsymbol{g}$ respectively in our analysis.

The optimization problem for network pruning using both the first and second-order terms can be depicted in Equation 6. Here, $\boldsymbol{E}$ is the error or loss function, $\boldsymbol{w}$ is the weight vector for the neural network, and $\delta \boldsymbol{w}$ is the change in the weight vector. Additionally, $I_m$ is the unit vector in weight space corresponding to the pruned weight $w_m$, $\mathbf{H} = \frac{\partial^2 \boldsymbol{E}}{\partial \boldsymbol{w}^2}$ denotes the Hessian Matrix, and the superscript $\top$ signifies vector transpose.

$$\min_{m} \left\{ \min_{\delta \boldsymbol{w}} \left( \left( \frac{\partial \boldsymbol{E}}{\partial \boldsymbol{w}} \right)^\top \cdot \delta \boldsymbol{w} + \frac{1}{2} \delta \boldsymbol{w}^\top \cdot \mathbf{H} \cdot \delta \boldsymbol{w} \right) \middle| I_m^\top \cdot \delta \boldsymbol{w} + w_m = 0 \right\} \quad (6)$$

By solving the optimization problem, we obtain the optimal change in error, $\delta \boldsymbol{E}_m$, for removing weight $w_m$ as shown in Equation 7. We have provided a detail analysis in Appendix G.

$$\delta \boldsymbol{E}_m = \frac{w_m^2}{2 \left( \mathbf{H}^{-1} \right)_{mm}} - \frac{w_m \left( \boldsymbol{g}^\top \cdot \mathbf{H}^{-1} \cdot I_m \right)}{\left( \mathbf{H}^{-1} \right)_{mm}} + \frac{\left( I_m^\top \cdot \mathbf{H}^{-1} \cdot \boldsymbol{g} \right)^2}{2 \left( \mathbf{H}^{-1} \right)_{mm}} - \frac{1}{2} \boldsymbol{g}^\top \cdot \mathbf{H}^{-1} \cdot \boldsymbol{g} \quad (7)$$

For the error, $\delta \boldsymbol{E}_m$, since the gradients are already small, we can consider the quadratic or square term of the gradient to be insignificant. Thus, ignoring the third and fourth terms, we have:

$$\delta \boldsymbol{E}_m = \frac{w_m^2}{2 \left( \mathbf{H}^{-1} \right)_{mm}} - \frac{w_m \left( \boldsymbol{g}^\top \cdot \mathbf{H}^{-1} \cdot I_m \right)}{\left( \mathbf{H}^{-1} \right)_{mm}} \quad (8)$$

To compute the Hessian matrix, we draw upon the Optimal Brain Compression method introduced in the work by Frantar & Alistarh (2022). This method optimizes Hessian computation by breaking down the global compression task into layer-specific sub-problems. This approach results in a closed-form solution for the Hessian, as expressed in Equation $\mathbf{H} = 2X^{\top}X$.

Following Optimal Brain Damage (LeCun et al., 1989), we introduce a simplifying assumption wherein we restrict our focus to the diagonal elements of the Hessian matrix. This results in $\mathbf{H} = 2*$ $\mathrm{diag}\left(\left\{\|\boldsymbol{x}_j\|_2^2, 1 \leq j \leq n\right\}\right)$. Here $\boldsymbol{x}_j$ is the tensor corresponding to component $j$ of the activation tensor across samples, and the variable $n$ represents the total number of components within the activation tensor for the respective layer. So, the first term of Equation 8 transforms into:

$$\frac{w_m^2}{2\left(\mathbf{H}^{-1}\right)_{mm}} = w_m^2 \left\|\boldsymbol{x}_m\right\|_2^2 \tag{9}$$

Since we are considering only the diagonal elements of Hessian $\mathbf{H}$. The second term in Equation 8 transforms as follows:

$$-\frac{w_m\left(\boldsymbol{g}^{\top} \cdot \mathbf{H}^{-1} \cdot I_m\right)}{\left(\mathbf{H}^{-1}\right)_{mm}} = -\frac{w_m g_m\left(\mathbf{H}^{-1}\right)_{mm}}{\left(\mathbf{H}^{-1}\right)_{mm}} = w_m(-g_m) \tag{10}$$

Thus, the final solution for the optimization problem in Equation 6 can be expressed as:

$$\delta\boldsymbol{E}_m = \left(w_m \left\|\boldsymbol{x}_m\right\|_2\right)^2 + w_m\left(-g_m\right) \tag{11}$$

Building upon the solution outlined in Equation 11, we conduct a series of experiments with different formulations of pruning metric in Section 3.4. Our investigation reveals that the pruning metric $\left(w_m \cdot \|x_m\|_2 + |w_m| \cdot g_m\right)$ yields the most favorable results. Here $g_m$ is the gradient magnitude obtain by either the $l_1$ or $l_2$ normalization across samples.

## 3 EXPERIMENTS

### 3.1 IMPLEMENTATION AND SETUP DETAILS

We conduct all our experiments using PyTorch (Paszke et al., 2017) for GBLM-Pruner. Experiments are performed with six models from the LLaMA-1 series (7B, 13B, 30B) (Touvron et al., 2023a) and the LLaMA-2 series (7B, 13B, 70B) (Touvron et al., 2023b). The Huggingface transformer library is used (Wolf et al., 2019) for handling models. The experiments are conducted on NVIDIA A100 GPUs with 40/80GB of memory. GBLM-Pruner requires calibration data for the computation of gradients and activations. Following previous works (Frantar et al., 2022; Frantar & Alistarh, 2023; Sun et al., 2023), we use 128 sequences with 2048-tokens randomly sampled from the first shard of the C4 (Raffel et al., 2019) training data as our calibration data. The gradients are computed with language modeling on the input sequence as the objective function. This represents the pretraining objective of the language models and remains agnostic to the downstream task the language models are used for. For scaling factor $\alpha$, we use a value of 100 after careful calibration ablation, as shown in Section 3.4.

**Baseline Approaches.** We compare our proposed method against three baselines: **(1)** magnitude pruning, **(2)** SparseGPT (Frantar & Alistarh, 2023), and **(3)** Wanda (Sun et al., 2023). Following Gale et al. (2019a) and Sanh et al. (2020b), we conduct a layer-wise comparison of model weights for magnitude pruning, subsequently removing those with smaller magnitudes. For both SparseGPT[1] and Wanda[2], we utilize their respective code implementation to obtain the pruned models.

**Evaluation.** We assess the performance of the pruned models using two distinct metrics: **(1)** Perplexity and **(2)** Zero-shot Evaluation on the Harness Benchmark (Gao et al., 2021). Perplexity is a well-established metric (Dettmers & Zettlemoyer, 2022; Yao et al., 2022; Frantar & Alistarh, 2022; Sun et al., 2023; Frantar & Alistarh, 2023) and provides stable and reliable results. The Zero-shot Harness evaluation, although known to be relatively noisy, offers a more readily interpretable assessment of model performance.

---

[1] https://github.com/IST-DASLab/sparsegpt
[2] https://github.com/locuslab/wanda

**Sparsity and Pruning Granularity.** Following recent methods (Frantar & Alistarh, 2023; Sanh et al., 2020b), GBLM-Pruner prunes the linear layers of LLMs uniformly except for the embedding layer and the final classification head. In addition to unstructured pruning, we also position GBLM-Pruner in comparison to other baselines, exploring more rigorous

Table 1: Pruning granularity for GBLM-Pruner.

| Pruning Granularity | Perplexity |
|---|---|
| layer | 7.45 |
| input,1 | 10.16 |
| input,128 | 7.64 |
| output,1 | **6.86** |
| output,128 | 7.47 |

yet hardware-accommodating 2:4 and 4:8 semi-structured sparsity patterns. We experiment with five different pruning configurations, as shown in Table 1. Our findings indicate that the (output,1) configuration yields the most favorable results, prompting its adoption as the standard for all our experiments.

## 3.2 PERPLEXITY EVALUATION

For all the methods under consideration, we report the perplexity evaluated on WikiText (Merity et al., 2016) validation data for both unstructured and semi-structured N:M sparsity pruning in Table 2. For unstructured pruning, GBLM-Pruner with $\ell_1$ norm outperforms both Wanda and reconstruction-based SparseGPT significantly across both LLaMA-1 and LLaMA-2 models.

However, the N:M sparsity pruning is restrictive by definition, especially 2:4 sparsity, which imposes greater constraints and results in a noticeable decrease in perplexity compared to unstructured pruning. As shown in Table 2, we can observe SparseGPT seems to perform better than both GBLM-Pruner and Wanda in the case of 2:4 sparsity pruning. Conversely, for 4:8 sparsity pruning, GBLM-Pruner outperforms other baselines for most of models, especially for the larger models.

Table 2: WikiText perplexity of pruning methods for LLaMA 1 and LLaMA 2 family of models.

| Method | Sparsity | LLaMA-2 | | | LLaMA-1 | | |
|---|---|---|---|---|---|---|---|
| | | 7B | 13B | 70B | 7B | 13B | 30B |
| None | 0 | 5.47 | 4.88 | 3.32 | 5.68 | 5.09 | 4.10 |
| Magnitude | 0.5 | 16.03 | 6.83 | 5.36 | 17.29 | 20.21 | 7.54 |
| SparseGPT | 0.5 | 7.00 | 6.03 | 4.25 | 7.22 | 6.19 | 5.32 |
| Wanda | 0.5 | 6.92 | 5.97 | 4.22 | 7.26 | 6.15 | 5.24 |
| GBLM-Pruner$_{\ell_1}$ | 0.5 | **6.86** | **5.88** | **4.17** | **7.15** | **6.11** | **5.18** |
| Magnitude | 2:4 | 37.77 | 8.89 | 6.76 | 42.54 | 18.36 | 9.11 |
| SparseGPT | 2:4 | **10.82** | **8.75** | 5.68 | **10.88** | **9.06** | 7.12 |
| Wanda | 2:4 | 12.11 | 9.00 | 5.48 | 11.53 | 9.59 | 6.90 |
| GBLM-Pruner$_{\ell_1}$ | 2:4 | 11.91 | 8.80 | **5.47** | 11.33 | 9.16 | **6.87** |
| Magnitude | 4:8 | 15.91 | 7.32 | 5.89 | 16.83 | 13.87 | 7.62 |
| SparseGPT | 4:8 | **8.46** | 7.01 | 4.91 | **8.45** | 7.44 | 6.18 |
| Wanda | 4:8 | 8.60 | 7.00 | 4.77 | 8.57 | 7.41 | 5.97 |
| GBLM-Pruner$_{\ell_1}$ | 4:8 | 8.63 | **6.90** | **4.72** | 8.48 | **7.26** | **5.89** |

## 3.3 ZERO-SHOT TASKS

In addition to our perplexity evaluations, we further assess the performance of our method across six Zero-shot common-sense tasks included in the Eleuther AI lm-evaluation-harness benchmark (Gao et al., 2021): BoolQ (Clark et al., 2019), RTE (Wang et al., 2018), HellaSwag (Zellers et al., 2019), WinoGrande (Sakaguchi et al., 2019), ARC-easy (Clark et al., 2018), and OBQA (Mihaylov et al., 2018). As noted by earlier work (Dettmers & Zettlemoyer, 2022; Frantar & Alistarh, 2023), zero-shot evaluation on these tasks is known to be noisy but aggregate performance across multiple tasks enhances interpretability.

Our comprehensive results for these tasks are presented in Table 3, where models are pruned to 50% unstructured sparsity. Notably, while our proposed GBLM-Pruner outperforms both Wanda and SparseGPT in terms of perplexity, a consistent trend is not observed across all the individual tasks, which aligns with existing literature (Frantar & Alistarh, 2023; Dettmers & Zettlemoyer, 2022). However, the mean accuracy across all six tasks surpasses the performance of both SparseGPT and Wanda for most of the models. This observation aligns with our findings from the perplexity evaluation, suggesting the robustness and effectiveness of our approach.

Table 3: Zero-Shot harness evaluation on 50% unstructured sparsity pruned models.

| Models | Method | BoolQ | RTE | HellaSwag | WinoGrande | ARC-e | OBQA | Mean |
|---|---|---|---|---|---|---|---|---|
| | Dense | 75.11 | 66.43 | 76.21 | 69.85 | 72.81 | 44.40 | 67.47 |
| | Mag | 54.65 | 54.15 | 60.90 | 59.43 | 54.38 | 35.20 | 53.12 |
| LLaMA-1-7B | SparseGPT | 72.87 | 53.07 | 69.77 | 67.88 | 66.46 | 40.60 | 61.77 |
| | Wanda | 71.25 | 54.87 | 70.12 | 66.06 | 65.11 | 39.60 | 61.17 |
| | Ours | **73.43** | **59.93** | **70.29** | **67.40** | **65.99** | **41.40** | **63.07** |
| | Dense | 77.98 | 70.40 | 79.07 | 72.77 | 74.75 | 44.80 | 69.96 |
| | Mag | 54.95 | 50.90 | 59.69 | 63.54 | 54.25 | 39.80 | 53.86 |
| LLaMA-1-13B | SparseGPT | 76.67 | 63.18 | 74.09 | 71.59 | 68.48 | 43.60 | 66.27 |
| | Wanda | 76.02 | 63.18 | 74.80 | **71.90** | 69.82 | 43.00 | 66.45 |
| | Ours | **76.61** | 63.18 | **74.90** | 71.67 | **70.37** | **43.20** | **66.65** |
| | Dense | 82.72 | 66.79 | 82.62 | 75.77 | 78.91 | 48.20 | 72.50 |
| | Mag | 64.25 | 49.82 | 67.29 | 66.61 | 70.71 | 41.20 | 59.98 |
| LLaMA-1-30B | SparseGPT | 82.91 | 55.96 | 79.31 | 74.27 | 77.53 | 46.00 | 69.33 |
| | Wanda | 81.71 | 65.34 | 79.91 | 73.56 | **78.11** | **46.40** | 70.84 |
| | Ours | **82.69** | **67.15** | **80.23** | 73.95 | 76.98 | 46.00 | **71.17** |

## 3.4 ABLATION STUDY

**Importance of Gradient**. To emphasize the role of gradient, we perform an ablation experiment as shown in Table 4, wherein we only consider the Gradient-Weight term of the `GBLM-Pruner` pruning metric.

Our experiments show a substantial enhancement over magnitude-based pruning when utilizing gradients solely with weights, evident in both LLaMA-2 7B and 13B models. Additionally, the performance of our metric closely aligns with that of Wanda and SparseGPT for LLaMA-2 13B model.

Table 4: Gradient-Weight based pruning metric.

| Method | Sparsity | 7B | 13B |
|---|---|---|---|
| Magnitude | 0.5 | 16.03 | 6.83 |
| SparseGPT | 0.5 | 7.00 | 6.03 |
| Wanda | 0.5 | **6.92** | 5.97 |
| $\|\mathbf{W}\| \cdot \|\mathbf{G}\|_1$ (Ours) | 0.5 | 7.17 | 6.15 |
| $\|\mathbf{W}\| \cdot \|\mathbf{G}\|_2$ (Ours) | 0.5 | 7.09 | **5.96** |

**Pruning Metric**. In Section 2.3, we revisited the OBS framework by incorporating the first order gradient which yields $\delta \boldsymbol{E}_m = (\boldsymbol{w}_m \|x_m\|_2)^2 + \boldsymbol{w}_m (-\boldsymbol{g}_m)$ as the pruning metric. To start with, we experiment with different ways of estimating the gradient magnitude from the calibration samples. We evaluated three methods: gradient accumulation, $\ell_1$ norm and $\ell_2$ norm applied to the gradient across calibration samples. For this experiment, we only utilize the pruning metric based on gradient alone with weight for better interpretability. From our experiment, we observe that gradient accumulation yields the least favorable results as depicted in Table 5. For deeper understanding, we compared the pruning pattern of gradient accumulation with $\ell_1$ and $\ell_2$ norm which shows that gradient accumulation gives a noisy estimate of the gradient magnitude while $\ell_1$ and $\ell_2$ norm reveals more structured patterns. A comparison between gradient accumulation and $\ell_1$ norm-based aggregation is shown in Figure 4. Based on this, we adopt $\ell_1$ and $\ell_2$ norm-based gradient estimation for subsequent analysis.

Subsequently, based on our theoretical pruning metric $\delta \boldsymbol{E}_m$, we experiment with two different ways of coupling the activations and gradients as shown in Table 5. We observe that in the case of $(\|\mathbf{W}\| \cdot \|\mathbf{X}\|_2)^2 - \|\mathbf{W}\| \cdot \|\mathbf{G}\|_p$ the pruning metric is completely disrupted. While for $(\|\mathbf{W}\| \cdot \|\mathbf{X}\|_2)^2 + \|\mathbf{W}\| \cdot \|\mathbf{G}\|_p$ gradient and activations complements each other and brings out the best performance. But, upon closer examination, we observe that the square of the first activation term sig-

Table 5: Pruning metric on weight, gradient, activation.

| Method | Sparsity | Perplexity |
|---|---|---|
| $\|\mathbf{W}\| \cdot \|\mathbf{G}_{acc}\|$ | 0.5 | 119.72 |
| $\|\mathbf{W}\| \cdot \|\mathbf{G}\|_1$ | 0.5 | 7.17 |
| $\|\mathbf{W}\| \cdot \|\mathbf{G}\|_2$ | 0.5 | 7.09 |
| $(\|\mathbf{W}\| \cdot \|\mathbf{X}\|_2)^2 + \alpha \cdot \|\mathbf{W}\| \cdot \|\mathbf{G}\|_1$ | 0.5 | 6.90 |
| $(\|\mathbf{W}\| \cdot \|\mathbf{X}\|_2)^2 + \alpha \cdot \|\mathbf{W}\| \cdot \|\mathbf{G}\|_2$ | 0.5 | 6.88 |
| $(\|\mathbf{W}\| \cdot \|\mathbf{X}\|_2)^2 - \alpha \cdot \|\mathbf{W}\| \cdot \|\mathbf{G}\|_1$ | 0.5 | 9743.65 |
| $(\|\mathbf{W}\| \cdot \|\mathbf{X}\|_2)^2 - \alpha \cdot \|\mathbf{W}\| \cdot \|\mathbf{G}\|_2$ | 0.5 | 9377.00 |
| $\|\mathbf{W}\| \cdot \|\mathbf{X}\|_2 + \alpha \cdot \|\mathbf{W}\| \cdot \|\mathbf{G}\|_1$ | 0.5 | **6.86** |
| $\|\mathbf{W}\| \cdot \|\mathbf{X}\|_2 + \alpha \cdot \|\mathbf{W}\| \cdot \|\mathbf{G}\|_2$ | 0.5 | 6.89 |

nificantly outweighs the contribution of the second term involving gradients. Consequently, we remove the square factor from the first term and add a scaling factor denoted as $\alpha$ to the second gradient term, resulting in the formulation of our final pruning metric as $\|\mathbf{W}\| \cdot \|\mathbf{X}\|_2 + \alpha \cdot \|\mathbf{W}\| \cdot \|\mathbf{G}\|_p$. This pruning metric with $\ell_1$ norm-based gradient aggregation gives the best result for unstructured pruning across all models. We also conduct experiments to calibrate the scal-

ing factor $\alpha$ as shown in Table 6. We vary the scaling factor and examine how the LLaMA-2-7B pruned model perplexity changes. For a scaling factor is equal to 100, we get the best perplexity.

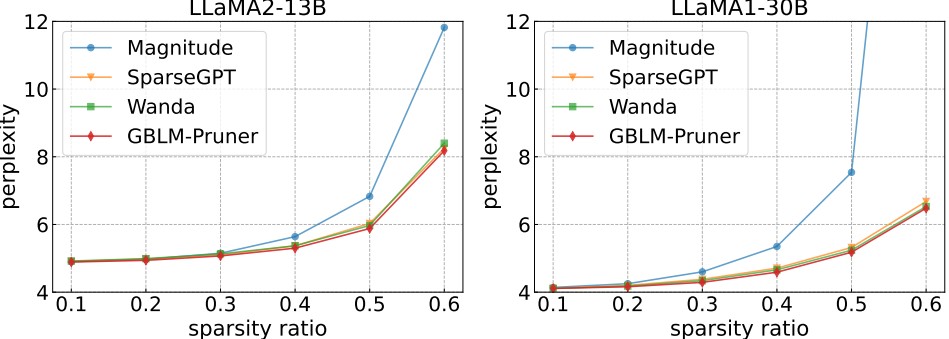

Figure 2: Sparsity variation results for a large and a small model where we compare the performance of our method against other baseline methods.

**Sparsity Variation**. The objective of this ablation is to assess the robustness of our method across varying sparsity. For this, we compare the perplexity of the unstructured pruned model obtained by `GBLM-Pruner` to that of Wanda, SparseGPT, and magnitude pruning. We consider two distinct model sizes: a smaller LLaMA-2 13B model and a larger LLaMA-1 30B model, each is subjected to different degrees of sparsity. The results are shown in Figure 2. From the figure, it is evident that `GBLM-Pruner` exhibits a similar trend to SparseGPT and Wanda, showing a decline in performance as sparsity increases. However, `GBLM-Pruner` consistently outperforms all other baseline methods across various levels of sparsity for both models.

Table 6: Ablation of scaling factor.

| Scaling Factor, ($\alpha$) | Perplexity |
|---|---|
| 0.001 | 6.920 |
| 0.01 | 6.919 |
| 0.1 | 6.921 |
| 1 | 6.912 |
| 10 | 6.890 |
| 100 | **6.858** |
| 1000 | 6.902 |
| 10000 | 6.926 |
| 100000 | 6.952 |

**Dependence on Calibration Sample**. `GBLM-Pruner` uses a set of calibration samples to calculate gradients and activations for the pruning metric. To understand the robustness of the pruned model to the calibration set, we conduct two ablations:

**(1)** Robustness to calibration set: For this ablation, we randomly sampled 5 different calibration sample sets with 128 samples each and pruned the LLaMA-2 7B model to 0.5 sparsity using `GBLM-Pruner`. The resultant pruned models have perplexities: 6.86, 6.87, 6.89, 6.86, and 6.87 respectively. The average perplexity is 6.87 which is close to our reported perplexity in Table 2.

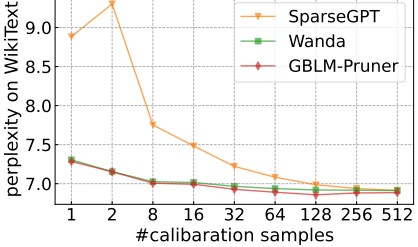

Figure 3: Robustness to calibration samples.

**(2)** Number of samples in the calibration set: In this experiment, we want to assess the influence of the calibration set size on the performance of `GBLM-Pruner`. For this, we prune the LLaMA-2 7B model using various calibration sets with the number of samples ranging from 1 to 512. The results are reported in Figure 3. From the figure, we can observe that in contrast to SparseGPT, our method exhibits a relatively lower sensitivity to variations in the number of calibration samples.

### 3.5 Visualization of Pruned Pattern

The visualization of learned pruning pattern is illustrated in Figure 4. To elaborate, on the left is the mask that is acquired by eliminating 50% of gradient from the summation-aggregated gradient tensor of the first layer's key projection, on the right is the mask that is derived by discarding 50% of the gradient from the $\ell_1$-norm-aggregated gradient tensor of the same layer's key projection. Within each subfigure, the x-axis represents the input dimension and

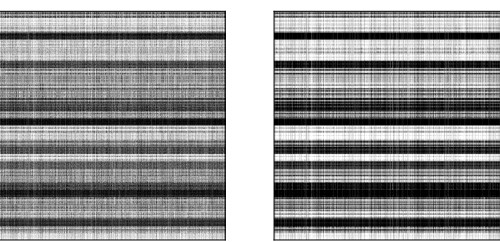

Figure 4: Illustration of learned pruning pattern.

the y-axis symbolizes the output dimension. The mask derived from the summation-accumulated gradient tensor tends to be noisy, in contrast, the one obtained through the $\ell_1$ norm accumulated gradient tensor appears to be more refined and distinct. After the integration of gradients, the method of unstructured pruning tends to unveil certain structural patterns following the pruning process. This reflects the inherent geometric interdependence found in the parameter structure of the LLMs, which is highly aligned with the structure of gradients.

### 3.6 VISION TRANSFORMERS

To assess the generalizability of our method across models with different input modalities, we conduct experiments on the ViT-B model. We compare the performance of the pruned model obtained using `GBLM-Pruner` with those obtained through magnitude pruning and the Wanda method. We use 4,096 random

Table 7: ViT-B model pruning.

| Sparsity | Wanda | Magnitude | Ours $\ell_1$ | Ours $\ell_2$ |
|----------|-------|-----------|---------------|---------------|
| 0 | 75.40 | 75.40 | 75.40 | 75.40 |
| 0.5 | 64.54 | 59.48 | 64.64 | **64.86** |
| 0.6 | 43.65 | 29.98 | 44.15 | **44.23** |
| 0.7 | 7.92 | 1.88 | **8.89** | 8.02 |
| 0.8 | 0.20 | 0.18 | **0.32** | 0.24 |

samples from ImageNet-1k training set as our calibration data, and subsequently, we evaluate the pruned models on the standard ImageNet-1k classification task. The results of these evaluations are presented in Table 7. From the table, it is evident that our model outperforms both the Wanda method and magnitude pruning, particularly when dealing with higher levels of sparsity.

## 4 RELATED WORK

Large Language Models (LLMs) based on transformer architecture (Vaswani et al., 2017) have ushered in a transformative era in the realm of natural language processing, achieving outstanding success. Their consistent and remarkable performance spans a wide array of tasks (Brown et al., 2020b; Chung et al., 2022; Touvron et al., 2023a;b; Rozière et al., 2023; OpenAI, 2023; Anil et al., 2023). For a long time, pruning has been identified as a powerful technique for reducing the size or complexity of a model by removing unnecessary or redundant components (LeCun et al., 1989; Hassibi et al., 1993a). Pruning can be divided into structured and unstructured pruning. Structured pruning targets at removing a set of weights from a network at once such as channels or layers to reduce the model size and complexity while maintaining the network structure intact. In the realm of pruning LLMs for sparsity, several studies (Frantar & Alistarh, 2022; 2023; Sun et al., 2023) have been undertaken in this area. Our work also provides a unique angle from *gradient* along this direction.

## 5 CONCLUSION

We have presented a gradient-based pruning approach `GBLM-Pruner` for large language models (LLMs). Our approach performs in a training-free manner and applies gradient-based statistical magnitude to discern and selectively prune the model's parameters, maintaining unstructured sparsity throughout the model, thus enabling substantial reductions in model size while preserving the model's predictive accuracy. The proposed approach has surpassed all previous LLM pruning methods in terms of perplexity, zero-shot performance and interpretability, marking a pivotal advancement in the field. We also provided theoretical analyses on how gradients help identify the importance of weights in LLMs. The superior accuracy achieved by this approach not only highlights gradients' effectiveness that is supplemental to weights and activations but also establishes it as a benchmark in the realm of model pruning methods for LLMs. We hope the proposed approach could potentially facilitate the development of more efficient, scalable, and accessible language models, paving the way for new opportunities and applications across various domains leveraging *gradients*. The notable performance of this approach is indicative of the significant strides being made in optimizing LLMs and highlights the possibilities that lie ahead in the journey towards more sustainable and efficient language processing tasks.

## REPRODUCIBILITY STATEMENT

To ensure the reproducibility of our work, we provided our source code as a part of our supplementary submission. Further information on our experimental setup including details of datasets used and computational requirements in Section 3.1 and the appendix.

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

APPENDIX

## A   BASELINES

We compare our proposed method against three pruning baselines:

- **Magnitude pruning**: Magnitude pruning (Han et al., 2015b) is a simple and scalable pruning method where the importance of LLM weights is decided based on the absolute value of their magnitude. Following Gale et al. (2019a) and Sanh et al. (2020b), we conduct a layer-wise comparison of model weights, subsequently removing those with smaller magnitudes.

- **SparseGPT**: SparseGPT (Frantar & Alistarh, 2023) is based on the second-order Optimal Brain Surgeon framework (Hassibi et al., 1993a). It optimizes the accurate Optimal Brain Surgeon framework and introduces the first accurate one-shot pruning method that works efficiently at the scale of billions of parameters.

- **Wanda**: Wanda (Sun et al., 2023) proposed a simple pruning metric and showed the importance of activations in addition to weight magnitude while selecting weights for pruning. Unlike previous algorithms, it does not require any weight update of the remaining weights.

## B   EVALUATION METRIC

Perplexity and Zero-shot Evaluation on Harness are two well-established metric for evaluating compressed models:

- **Perplexity**: Following previous work on model compression both in case of quantization (Dettmers & Zettlemoyer, 2022; Yao et al., 2022) and pruning (Frantar & Alistarh, 2022; Sun et al., 2023; Frantar & Alistarh, 2023) we used perplexity as an evaluation metric to compare the pruned models. Perplexity is a stable, robust and challenging metric that is suited for evaluating the accuracy of compression methods. We used the WikiText (Merity et al., 2016) validation set for computing perplexity.

- **Zero-Shot Evaluation on Harness Benchmarks**: To complement perplexity, we provided the evaluation of the pruned model on the publicly available Eleuther AI LM Harness benchmark (Gao et al., 2021) for additional interpretability. We conducted evaluations on five standard common-sense reasoning tasks, including RTE (Wang et al., 2018), HellaSwag (Zellers et al., 2019), WinoGrande (Sakaguchi et al., 2019), ARC-easy (Clark et al., 2018), OBQA (Mihaylov et al., 2018) and the BoolQ (Clark et al., 2019) reading comprehension task. Our evaluation primarily centers on assessing the pruned models' accuracy in comparison to the dense baseline, rather than emphasizing absolute numerical values.

## C   PRUNING GRANULARITY

Pruning Granularity plays a pivotal role even in unstructured pruning. For `GBLM-Pruner`, we have experimented with 5 different pruning granularity:

- `Layer-wise`: With layer-wise pruning, weights within same layer are compared for puning.

- `(input, 1)`: For (input,1), weights connected within an input channel are grouped together for comparison.

- `(output, 1)`: Similarly in this approach, weights connected within an output channel are grouped together for comparison.

- `(input, 128)`: This pruning granularity involves forming blocks of 128 input channels, and weights within each block are compared for pruning.

- `(input, 128)`: Similar to (input,128), here blocks of 128 channels are formed along the output dimension for pruning.

## D    LLaMA-Chat Models

The LLaMA-2 series of models also includes fine-tuned chat versions. We sought to assess the generalization of our method to these chat models, specifically focusing on LLaMA-2-chat-7B and LLaMA-2-chat-13B as representative models. Similar to the pretrained LLaMA-2 series, our calibration data consisted of 128 samples, each comprising 2048 tokens from the C4 dataset. For evaluation purposes, we employed the Wiki-Text validation set.

Our approach to pruning was consistent with that applied to the pretrained LLaMA-2 models. We uniformly pruned every linear layer, except for the initial embedding layer and the final classification layer. We compare every weight of the linear layer on per output basis where pruning metric is compared within the output neuron.

The results are presented in Table 8. Examining the table, we can discern that our method consistently delivers superior performance, particularly evident in unstructured pruning. When it comes to N:M sparsity pruning, although SparseGPT achieves the lowest perplexity, our pruning metric significantly outperforms Wanda by a substantial margin.

Table 8: WikiText validation perplexity of different pruning methods for LLaMA 2 chat models.

| Method | Sparsity | LLaMA-2-7B-chat | LLaMA-13B-chat |
|---|---|---|---|
| None | 0 | 7.08 | 6.11 |
| Magnitude | 0.5 | 22.82 | 8.49 |
| Sparsegpt | 0.5 | 8.66 | 7.26 |
| Wanda | 0.5 | 8.78 | 7.50 |
| GBLM-Pruner$_{\ell 2}$ | 0.5 | 8.52 | 7.27 |
| GBLM-Pruner$_{\ell 1}$ | 0.5 | **8.40** | **7.10** |
| Magnitude | 2:4 | 45.95 | 11.14 |
| Sparsegpt | 2:4 | **12.19** | **9.37** |
| Wanda | 2:4 | 14.45 | 10.25 |
| GBLM-Pruner$_{\ell 2}$ | 2:4 | 13.74 | 9.85 |
| GBLM-Pruner$_{\ell 1}$ | 2:4 | 13.92 | 9.66 |
| Magnitude | 4:8 | 22.57 | 9.80 |
| Sparsegpt | 4:8 | **10.02** | **8.01** |
| Wanda | 4:8 | 10.86 | 8.56 |
| GBLM-Pruner$_{\ell 2}$ | 4:8 | 10.45 | 8.26 |
| GBLM-Pruner$_{\ell 1}$ | 4:8 | 10.46 | 8.10 |

## E    OBS Weight Update

In this study, our objective was to assess whether the OBS (Optimal Brain Surgeon) weight update method enhances the performance of our pruned model. We implemented the OBS weight update using the efficient approach proposed by SparseGPT (Frantar & Alistarh, 2023).

The results, presented in Table 9, indicate that the OBS weight update does not lead to an improvement in the performance of our pruned model

Table 9: OBS weight update.

| Method | Datasplit | Weight Update | |
|---|---|---|---|
| | | no | yes |
| Magnitude | Calib | 18.14 | 12.93 |
| | Valid | 17.29 | 12.55 |
| Wanda | Calib | 7.52 | 7.61 |
| | Valid | 7.26 | 7.36 |
| Ours | Calib | 7.54 | 7.64 |
| | Valid | 7.26 | 7.39 |

## F  CORRELATIONS OF WEIGHTS, ACTIVATIONS AND GRADIENTS.

This section discusses an intuitive explanation of why gradient is essential. Weights are parameters in LLMs that are learned during the training process to minimize the loss function. They are fundamental in determining the strength of the connection between two neurons and subsequently the output of the network. Gradients of the loss with respect to weights, computed using an optimization algorithm like SGD (Ruder, 2016), are central to the learning process as they guide the updates made to the weights during training. On the otherhand, activations are the outputs of the neurons, typically computed as a weighted sum of inputs passed through an activation function. The activations are intrinsically impacted by the weights thus weight augmented with activation serves as a redundant indicator of weight importance. However, gradient being the guiding signal for the learning process serves as a valuable indicator by signalling the sensitivity of the loss to weight change and thus the importance of the weight in the pruning process.

## G  OPTIMAL BRAIN SURGEON CONSIDERING GRADIENT

As a part of the theoretical justification for our proposed gradient-based metric, we revisited and redefined the OBS framework by incorporating considerations of the gradient information. The complete derivation of this process is meticulously presented within this section.

The Taylor Series expansion of the error with respect to weight is:

$$\delta \boldsymbol{E} = \left(\frac{\partial \boldsymbol{E}}{\partial \boldsymbol{w}}\right)^\top \cdot \delta \boldsymbol{w} + \frac{1}{2}\delta \boldsymbol{w}^\top \cdot \mathbf{H} \cdot \delta \boldsymbol{w} + \mathcal{O}(||\delta \boldsymbol{w}||^3) \tag{12}$$

where $\boldsymbol{E}$ is the error or loss function and $\boldsymbol{w}$ is the weight vector for the neural network. The symbol $\mathbf{H} = \frac{\partial^2 \boldsymbol{E}}{\partial \boldsymbol{w}^2}$ denotes the Hessian Matrix, and the superscript $\top$ signifies vector transpose. Based on this we formulate the optimization problem for network pruning using both the first and second-order terms as depicted in Equation 13. Here, $\boldsymbol{w}_m$ is the pruned weight, $\delta \boldsymbol{w}$ is the change in weight magnitude for $\boldsymbol{w}_m$ and $I_m$ is the unit vector in weight space corresponding to weight $\boldsymbol{w}_m$.

$$\min_q \left\{ \min_{\delta \boldsymbol{w}} \left( \left(\frac{\partial \boldsymbol{E}}{\partial \boldsymbol{w}}\right)^\top \cdot \delta \boldsymbol{w} + \frac{1}{2}\delta \boldsymbol{w}^\top \cdot \mathbf{H} \cdot \delta \boldsymbol{w} \right) \middle| I_m^\top \cdot \delta \boldsymbol{w} + w_m = 0 \right\} \tag{13}$$

The Lagrangian formulation of the optimization problem is:

$$\mathcal{L} = \boldsymbol{g}^\top \cdot \delta \boldsymbol{w} + \frac{1}{2}\delta \boldsymbol{w}^\top \cdot \mathbf{H} \cdot \delta \boldsymbol{w} + \lambda \left(I_m^\top \cdot \delta \boldsymbol{w} + w_m\right) \tag{14}$$

Now, differentiating Equation 14 w.r.t $\lambda$

$$I_m^\top \cdot \delta \boldsymbol{w} + w_m = 0 \tag{15}$$

Differentiating w.r.t $\delta \boldsymbol{w}$

$$\boldsymbol{g} + \mathbf{H} \cdot \delta \boldsymbol{w} + \lambda I_m = 0$$
$$\Rightarrow \delta \boldsymbol{w} = -\mathbf{H}^{-1} \cdot (\lambda I_m + \boldsymbol{g}) \tag{16}$$

From 15 and 16, we have

$$I_m^\top \left(-\mathbf{H}^{-1} \cdot (\lambda I_m + \boldsymbol{g})\right) + w_m = 0$$
$$\Rightarrow -\lambda \left(\boldsymbol{H}^{-1}\right)_{mm} - I_m^\top \cdot \mathbf{H}^{-1} \cdot \boldsymbol{g} + w_m = 0$$
$$\Rightarrow \lambda = \frac{w_m - I_m^\top \cdot \mathbf{H}^{-1} \cdot \boldsymbol{g}}{\left(\mathbf{H}^{-1}\right)_{qq}} \tag{17}$$

From 16 and 17, we get the optimal weight change $\delta \boldsymbol{w}$ as:

$$
\begin{aligned}
\delta \boldsymbol{w} &= -\mathbf{H}^{-1} \cdot \left( \frac{w_m - I_m^\top \cdot \mathbf{H}^{-1} \cdot \boldsymbol{g}}{\left(\mathbf{H}^{-1}\right)_{mm}} \cdot I_m + \boldsymbol{g} \right) \\
&= -\frac{w_m}{\left(\mathbf{H}^{-1}\right)_{mm}} \mathbf{H}^{-1} \cdot I_m + \frac{I_m^\top \cdot \mathbf{H}^{-1} \cdot g}{\left(\mathbf{H}^{-1}\right)_{mm}} \mathbf{H}^{-1} \cdot I_m - \mathbf{H}^{-1} \cdot \boldsymbol{g}
\end{aligned}
\tag{18}
$$

The increase in error on changing weight $w_m$ by $\delta \boldsymbol{w}$ is:

$$
\delta \boldsymbol{E}_m = \boldsymbol{g}^\top \cdot \delta \boldsymbol{w} + \frac{1}{2} \delta \boldsymbol{w}^\top \cdot \mathbf{H} \cdot \delta \boldsymbol{w}
\tag{19}
$$

Substituting the optimal value of $\delta \boldsymbol{w}$ in Equation 19 gives:

$$
\delta \boldsymbol{E}_m = \frac{w_m^2}{2 \left(\mathbf{H}^{-1}\right)_{mm}} - \frac{w_m \left(\boldsymbol{g}^\top \cdot \mathbf{H}^{-1} \cdot I_m\right)}{\left(\mathbf{H}^{-1}\right)_{mm}} + \frac{\left(I_q^\top \cdot \mathbf{H}^{-1} \cdot \boldsymbol{g}\right)^2}{2 \left(\mathbf{H}^{-1}\right)_{mm}} - \frac{1}{2} \boldsymbol{g}^\top \cdot \mathbf{H}^{-1} \cdot \boldsymbol{g}
\tag{20}
$$

## H  DIFFERENT PRUNING METRIC

In the ablation Section 3.4, we present an analysis of our pruning metric. Table 10 enumerates all the pruning metrics we explored and serves as a comprehensive consolidation of our study.

Table 10: Pruning metric.

| Method | Sparsity | Perplexity | Method | Sparsity | Perplexity |
|---|---|---|---|---|---|
| $\|\mathbf{W}\| \cdot \|\mathbf{G}_{acc}\|$ | 0.5 | 119.72 | $(\|\mathbf{W}\| \cdot \|\mathbf{X}\|_2)^2 + \alpha \cdot \mathbf{W} \cdot \mathbf{G}_{acc}$ | 0.5 | 7.04 |
| $\|\mathbf{W}\| \cdot \|\mathbf{G}\|_1$ | 0.5 | 7.17 | $(\|\mathbf{W}\| \cdot \|\mathbf{X}\|_2)^2 + \alpha \cdot \mathbf{W} \cdot \|\mathbf{G}\|_1$ | 0.5 | 180490.19 |
| $\|\mathbf{W}\| \cdot \|\mathbf{G}\|_2$ | 0.5 | 7.09 | $(\|\mathbf{W}\| \cdot \|\mathbf{X}\|_2)^2 + \alpha \cdot \mathbf{W} \cdot \|\mathbf{G}\|_2$ | 0.5 | 91781.49 |
| $\|\mathbf{W}\| \cdot \|\mathbf{X}\|_2 \cdot \|\mathbf{G}_{acc}\|$ | 0.5 | 69.59 | $(\|\mathbf{W}\| \cdot \|\mathbf{X}\|_2)^2 - \alpha \cdot \mathbf{W} \cdot \mathbf{G}_{acc}$ | 0.5 | 7.14 |
| $\|\mathbf{W}\| \cdot \|\mathbf{X}\|_2 \cdot \|\mathbf{G}\|_1$ | 0.5 | 7.31 | $(\|\mathbf{W}\| \cdot \|\mathbf{X}\|_2)^2 - \alpha \cdot \mathbf{W} \cdot \|\mathbf{G}\|_1$ | 0.5 | 246846.28 |
| $\|\mathbf{W}\| \cdot \|\mathbf{X}\|_2 \cdot \|\mathbf{G}\|_2$ | 0.5 | 7.31 | $(\|\mathbf{W}\| \cdot \|\mathbf{X}\|_2)^2 - \alpha \cdot \mathbf{W} \cdot \|\mathbf{G}\|_2$ | 0.5 | 283620.75 |
| $\|\mathbf{W}\| \cdot \|\mathbf{X}\|_2 + \alpha \cdot \|\mathbf{W}\| \cdot \|\mathbf{G}_{acc}\|$ | 0.5 | 6.92 | $(\|\mathbf{W}\| \cdot \|\mathbf{X}\|_2)^2 + \alpha \cdot \|\mathbf{W}\| \cdot \|\mathbf{G}_{acc}\|$ | 0.5 | 6.91 |
| $\|\mathbf{W}\| \cdot \|\mathbf{X}\|_2 + \alpha \cdot \|\mathbf{W}\| \cdot \|\mathbf{G}\|_1$ | 0.5 | **6.86** | $(\|\mathbf{W}\| \cdot \|\mathbf{X}\|_2)^2 + \alpha \cdot \|\mathbf{W}\| \cdot \|\mathbf{G}\|_1$ | 0.5 | 6.90 |
| $\|\mathbf{W}\| \cdot \|\mathbf{X}\|_2 + \alpha \cdot \|\mathbf{W}\| \cdot \|\mathbf{G}\|_2$ | 0.5 | 6.89 | $(\|\mathbf{W}\| \cdot \|\mathbf{X}\|_2)^2 + \alpha \cdot \|\mathbf{W}\| \cdot \|\mathbf{G}\|_2$ | 0.5 | 6.88 |
| $\|\mathbf{W}\| \cdot \|\mathbf{X}\|_2 - \alpha \cdot \|\mathbf{W}\| \cdot \|\mathbf{G}_{acc}\|$ | 0.5 | 6.92 | $(\|\mathbf{W}\| \cdot \|\mathbf{X}\|_2)^2 - \alpha \cdot \|\mathbf{W}\| \cdot \|\mathbf{G}_{acc}\|$ | 0.5 | 6.94 |
| $\|\mathbf{W}\| \cdot \|\mathbf{X}\|_2 - \alpha \cdot \|\mathbf{W}\| \cdot \|\mathbf{G}\|_1$ | 0.5 | 1180.67 | $(\|\mathbf{W}\| \cdot \|\mathbf{X}\|_2)^2 - \alpha \cdot \|\mathbf{W}\| \cdot \|\mathbf{G}\|_1$ | 0.5 | 9743.65 |
| $\|\mathbf{W}\| \cdot \|\mathbf{X}\|_2 - \alpha \cdot \|\mathbf{W}\| \cdot \|\mathbf{G}\|_2$ | 0.5 | 7.10 | $(\|\mathbf{W}\| \cdot \|\mathbf{X}\|_2)^2 - \alpha \cdot \|\mathbf{W}\| \cdot \|\mathbf{G}\|_2$ | 0.5 | 9377.00 |

## I  ZERO-SHORT HARNESS EVALUATION ON LLaMA-2 MODELS

We have also conducted Zero-shot Harness evaluation on the LLaMA-2 series of model and the results are reported in Table 11.

Table 11: Zero-Shot harness evaluation on 50% unstructured sparsity pruned models.

| Models | Method | BoolQ | RTE | HellaSwag | WinoGrande | ARC-e | OBQA | Mean |
|---|---|---|---|---|---|---|---|---|
| | Dense | 80.55 | 65.34 | 79.38 | 72.22 | 77.44 | 45.20 | 70.02 |
| | Mag | 57.65 | 55.96 | 73.02 | 65.35 | 67.17 | 40.80 | 59.99 |
| LLaMA-2-13B | SparseGPT | **81.25** | **62.82** | 75.34 | 70.48 | 71.34 | 44.00 | 67.54 |
| | Wanda | 81.07 | 60.65 | **76.08** | 71.67 | 71.63 | 44.60 | 67.62 |
| | Ours | 80.89 | 60.65 | 76.03 | **71.82** | **72.26** | **44.80** | **67.74** |
| | Dense | 83.70 | 67.87 | 83.80 | 77.98 | 80.98 | 48.80 | 73.86 |
| | Mag | 71.11 | 60.65 | 79.31 | 73.56 | 74.71 | 44.20 | 67.25 |
| LLaMA-2-70B | SparseGPT | **85.26** | 70.76 | 81.43 | **78.30** | **79.84** | **48.40** | **74.00** |
| | Wanda | 83.27 | **71.84** | 81.49 | 77.35 | 78.62 | 47.60 | 73.36 |
| | Ours | 83.73 | 71.48 | **81.64** | 77.11 | 78.28 | 47.40 | 73.27 |

