# OpenReview forum: "Beyond Size: How Gradients Shape Pruning Decisions in Large Language Models"
_ICLR.cc/2024/Conference — Submitted to ICLR 2024_

### Official Review · Reviewer_dK5u · 2023-10-27

**Soundness:** 3 good
**Presentation:** 3 good
**Contribution:** 2 fair
**Rating:** 5
**Confidence:** 4

**Summary:**

This study introduces GBLM-Pruner, a gradient-based approach for the unstructured pruning of large language models (LLMs). The core idea of this research is centered around a Taylor expansion applied to the loss function. This method estimates the change in loss by employing a combination of first-order gradient and second-order approximation (OBD). Empirical evaluations using LLaMA and LLaMA-2 demonstrate that GBLM-Pruner outperforms other methods such as magnitude pruning, SparseGPT, and Wanda in terms of performance.

**Strengths:**

1. This paper highlights the significance of gradients in the pruning of large language models (LLMs). The author presents a Taylor-based approach to identify critical parameters, yielding favorable outcomes in comparison to earlier techniques.
2. The work sets robust benchmarks by contrasting the proposed methods with various existing baselines, offering valuable insights for the research community.

**Weaknesses:**

1. To my knowledge, SparseGPT is similarly a gradient-based approach, utilizing Taylor expansion and second-order Hessian for estimating parameter importance. In light of this, the contribution of the current work may appear somewhat constrained.
2. As depicted in Figure 2, SparseGPT, Wanda, and the newly introduced GBLM-Pruner exhibit closely comparable results, with only minor differences in Perplexity (PPL). There isn't compelling evidence to suggest that GBLM-Pruner significantly outperforms its predecessors.
3. It would be beneficial if the author could include data on the latency of the pruned LLMs, particularly in the context of 2:4 sparsity acceleration.

**Questions:**

Please refer to the weaknesses.

---

> ### Author Response · Authors · 2023-11-23
>
> Thank you for your valuable comments and feedback, they are very helpful for us to improve our paper. We are glad that you found our paper insightful. In the following, we have first stated your comments and provided our response.
>
>
>     Q1. To my knowledge, SparseGPT is similarly a gradient-based approach, utilizing Taylor expansion and second-order Hessian for estimating parameter importance. In light of this, the contribution of the current work may appear somewhat constrained.
>
>
> The reviewer mentions that SparseGPT is a gradient based method but this is a misunderstanding. SparseGPT is not a gradient-based method, instead it optimizes the hessian computation and weight update stage within the OBS framework for large-scale language models. It relies solely on weight and activation information to compute the pruning metric, expressed as $W_m[i, j] = \frac{|W[i, j]^2|}{\text{diag}(H^{-1})[j, j]}$. Here, the Hessian is denoted as $H = X^TX + \lambda I$, with $X$ representing the activation matrix, $I$ is the identity matrix, $W$ as the weight matrix and $\lambda$ is the damping term. We would like to point the reviewer to section 2.1 of the paper, where discussed prior method Wanda and SparseGPT in detail.
>
>     Q2. As depicted in Figure 2, SparseGPT, Wanda, and the newly introduced GBLM-Pruner exhibit closely comparable results, with only minor differences in Perplexity (PPL). There isn't compelling evidence to suggest that GBLM-Pruner significantly outperforms its predecessors.
>
> The reviewer notes that the GBLM-Pruner exhibit closely comparable results to SparseGPT and Wanda, with only minor differences in Perplexity (PPL). To address this concern, it's essential to highlight that the enhancement demonstrated by GBLM-Pruner in unstructured pruning is comparable to Wanda's performance over SparseGPT which is 0.08, 0.06 and 0.03 on LLaMA-2-7B/13B/70B respectively. Additionally, the benefit of use of gradient is evident for restrictive 2:4 semi-structured pruning. Although SparseGPT tends to exhibit a superior performance, but it is computationally expensive. Our proposed GBLM-Pruner outperforms Wanda by a appreciable margin, while maintaining a comparable computation time as shown in our following response. For instance, GBLM-Pruner outperforms Wanda by 0.53 perplexity score for LLaMA-2-7B and 0.20 perplexity score for  LLaMA-2-13B.
>
>     Q3. It would be beneficial if the author could include data on the latency of the pruned LLMs, particularly in the context of 2:4 sparsity acceleration.
>
> We appreciate the reviewer for highlighting this crucial question. As demonstrated by Wanda [1], 2:4 sparsity pruning yields a speedup of 1.24x in GPU for the LLaMA architecture. The table below provides an illustration of the evaluation of inference speedup for 2:4 sparsity on NVIDIA A6000 GPUs across various transformer layers of the LLaMA model.
>
> | LLaMA layer | Dense | 2:4  | Speedup |
> |----------|----------|----------|----------|
> | q/k/v/o_proj | 3.49   | 2.14   | 1.63x   |
> | up/gate_proj   | 9.82   | 6.10   | 1.61x  |
> | down_proj   | 9.92   | 6.45   | 1.54x  |
>
> [1] Sun, Mingjie et al. “A Simple and Effective Pruning Approach for Large Language Models.” ArXiv abs/2306.11695 (2023): n. pag.

---

### Official Review · Reviewer_gsUn · 2023-10-31

**Soundness:** 3 good
**Presentation:** 3 good
**Contribution:** 2 fair
**Rating:** 5
**Confidence:** 5

**Summary:**

This paper proposes to integrate the first-order gradient into the unstructured pruning of large language models and achieves superior performance compared to sparseGPT and Wanda.

**Strengths:**

1. A superior method compared to SparseGPT and Wanda on unstructured pruning of large language model
2. The authors have conducted extensive experiments to assess the method's effectiveness on LLaMa-1 and LLaMa-2. Additionally, the paper illustrates the impact of various gradient and activation combinations on the determination of parameter importance.
3. The paper is well-written, offering clarity and ease of understanding in its presentation.

**Weaknesses:**

1. The novelty of this method appears somewhat constrained. Utilizing the first-order gradient for determining parameter importance is a common approach in pruning techniques applied to CNN, BERT, and ViT. This technique is well-established within the realm of model pruning. Considering in some instances this method even falls short of those achieved by SparseGPT (e.g., 2:4 for LLaMA-1 and LLaMA-2), I cannot say the first-order gradient in pruning LLMs might be a major contribution.
2. This paper lacks experiments on different LLM families. Conducting trials with models like OPT, BLOOM, or other alternatives could provide valuable insights into the method's applicability and generalizability across various LLM families.
3. The paper doesn't provide details regarding the latency of the pruned model. In a study centered on LLM compression, including latency metrics is crucial since such information is highly important  to the readers to understand the efficiency of the pruned model.

**Questions:**

1. Could you specify the error function utilized for calculating gradients in your approach?
2. Have you conducted any latency experiments on the pruned model, particularly under the 2:4 or 4:8 configurations?
3. Is the calibration set employed for your methods and Wanda, SparseGPT identical?

---

> ### Author Response · Authors · 2023-11-23
>
> Thank you for your valuable comments and feedback, they are very helpful for us to improve our paper. We are glad that you found our paper well-written and easy to follow. In the following, we have first stated your comments and provided our response.
>
>     Q1. The novelty of this method appears somewhat constrained. Utilizing the first-order gradient for determining parameter importance is a common approach in pruning techniques applied to CNN, BERT, and ViT. This technique is well-established within the realm of model pruning. Considering in some instances this method even falls short of those achieved by SparseGPT (e.g., 2:4 for LLaMA-1 and LLaMA-2), I cannot say the first-order gradient in pruning LLMs might be a major contribution.
>
> The reviewer mentions that the novelty of GBLM-Pruner is constrained as there are existing studies such as [1,2,3,4] that have explored the utilization of gradient information for pruning neural networks. To address this concern, it is crucial to emphasize that our work is the first attempt to study the use of gradients for one-shot pruning of language models with billions of parameters while maintaining the zero-shot generalization capabilities of the language models to diverse downstream tasks. Meanwhile, [1,3] used gradient in the context of transfer learning to obtain a pruned model that preserves the accuracy of the finetuned downstream task. Unlike [1,3], where the gradients are obtained for the finetuning objective on a downstream task, our work uses the language modelling objective on a small set of calibration samples to obtain the gradients. Additionally, unlike [1,3], GBLM-Pruner requires no weight update to maintain the model performance. This makes our proposed method computationally efficient and applicable for large language models with billions of parameters like LLaMA-1-30B and LLaMA-2-70B.
>
> Furthermore, [2,4] ignore the first-order terms under the assumption that $\Delta L ≈ 0$ and cease to provide any useful information when higher-order terms are under consideration. Instead, they use gradient for optimizing the computation of the inverse Hessian by utilizing the empirical Fisher matrix in terms of gradients. We illustrated how gradient information can be integrated with higher order Hessian terms $(H = X^TX + \lambda I)$ or activations through the use of scaling term $\alpha$ to balance the magnitude difference between activations and gradients. This gives us superior performance than Wanda across different models. To underscore the significance of the scaling term, we have incorporated an ablation analysis in the table below. It presents the perplexity values obtained for the LLaMA-2-7B model across various $\alpha$ values.
>
> | Scaling term, $\alpha$ | Perplexity |
> |----------|----------|
> | 1 |  6.912 |
> | 10 |  6.890 |
> | 100   | 6.858 |
> | 1000   | 6.902 |
> | 10000   | 6.926 |
>
> The reviewer also points out that GBLM_Pruner falls short of SparseGPT for 2:4 sparsity. Although SparseGPT tends to exhibit a superior performance, it is computationally expensive. Our proposed GBLM-Pruner outperforms Wanda by an appreciable margin while maintaining a comparable computation time. For instance, GBLM-Pruner outperforms Wanda by 0.53 perplexity score for LLaMA-2-7B and 0.20 perplexity score for  LLaMA-2-13B.
>
> [1] Pruning convolutional neural networks for resource efficient inference, Molchanov et al.
>
> [2] WoodFisher: Efficient Second-Order Approximation for Neural Network Compression, Singh et al.
>
> [3] Movement Pruning: Adaptive Sparsity by Fine-Tuning, Sanh et al.
>
> [4] The Optimal BERT Surgeon: Scalable and Accurate Second-Order Pruning for Large Language Models, Kurtic et al.

---

> ### Author Response · Authors · 2023-11-23
>
> Q2. This paper lacks experiments on different LLM families. Conducting trials with models like OPT, BLOOM, or other alternatives could provide valuable insights into the method's applicability and generalizability across various LLM families.
>
>
> The reviewer notes that our experiments are confined to the LLaMA series of models. In order to showcase the generalizability of our methods across various LLM families, we conducted experiments with OPT-6.7B and OPT-13B models, as presented in the table below. Our findings indicate consistent results, demonstrating that GBLM-Pruner consistently offers appreciable improvements over Wanda.
>
> | method | ratio | Sparsity type |6.7B| 13B|
> |----------|----------|----------|----------|----------|
> |Dense| 0.0   | --  |10.86| 10.13 |
> |magnitude| 0.5   | unstructured  |968.72| 11644.11 |
> |Wanda| 0.5   | unstructured | 11.98 | 11.93  |
> |GBLM-Pruner $_{l1}$| 0.5   | unstructured  | **_11.89_**|**_11.53_** |
> |GBLM-Pruner $_{l2}$| 0.5   | unstructured  | 11.94 |11.85  ||
> |magnitude| 0.5   | 2:4  |264.15 | 484.78 |
> |Wanda| 0.5   | 2:4  |**_15.90_**| 15.56  |
> |GBLM-Pruner $_{l1}$| 0.5   | 2:4  | 16.04 |**_14.77_** |
> |GBLM-Pruner $_{l2}$| 0.5   | 2:4  | 15.94 | 15.39  |
> |magnitude| 0.5   | 4:8  |96.17| 449.66 |
> |Wanda| 0.5   | 4:8  |13.55 | 13.35  |
> |GBLM-Pruner $_{l1}$| 0.5   | 4:8  | **_13.48_** |**_12.74_** |
> |GBLM-Pruner $_{l2}$| 0.5   | 4:8  | 13.55 |13.26 |

---

> ### Author Response · Authors · 2023-11-23
>
> Q3. The paper doesn't provide details regarding the latency of the pruned model. In a study centered on LLM compression, including latency metrics is crucial since such information is highly important to the readers to understand the efficiency of the pruned model.
>
> We appreciate the reviewer for highlighting this crucial question. In response, we wish to underscore that unstructured pruning not only enhances CPU inference speed but also contributes to storage memory efficiency. Notably, a 2:4 sparsity ratio has been shown to yield a speedup of 1.24x in GPUfor the LLaMA architecture, as demonstrated by Wanda [1]. The table below provides an illustration of the evaluation of inference speedup for 2:4 sparsity on NVIDIA A6000 GPUs across various transformer layers of the LLaMA model.
>
> | LLaMA layer | Dense | 2:4  | Speedup |
> |----------|----------|----------|----------|
> | q/k/v/o_proj | 3.49   | 2.14   | 1.63x   |
> | up/gate_proj   | 9.82   | 6.10   | 1.61x  |
> | down_proj   | 9.92   | 6.45   | 1.54x  |
>
> It is to be noted that for 2:4 pruning, our proposed GBLM-Pruner outperforms Wanda by an appreciable margin. For instance, GBLM-Pruner outperforms Wanda by 0.53 perplexity score for LLaMA-2-7B and 0.20 perplexity score for LLaMA-2-13B.
>
> Q4. Could you specify the error function utilized for calculating gradients in your approach?
>
> We would like to point the reviewer to section 3.1 of the paper. We have stated that we use language modelling as error function for calculating gradients in our approach. This aligns with the pretraining objective that the language models is trained on and remains agnostic to the downstream task that the language model are used for.
>
> Q5. Is the calibration set employed for your methods and Wanda, SparseGPT identical?
>
> We would like to point the reviewer to section 3.1 of the paper. We have stated that following Wanda and SparseGPT, we use 128 sequences with 2048 tokens randomly sampled from the first shard of the C4 training data.
>
> [1] Sun, Mingjie et al. “A Simple and Effective Pruning Approach for Large Language Models.” ArXiv abs/2306.11695 (2023): n. pag.

---

### Official Review · Reviewer_B7C2 · 2023-10-31

**Soundness:** 2 fair
**Presentation:** 2 fair
**Contribution:** 2 fair
**Rating:** 3
**Confidence:** 4

**Summary:**

* The paper proposes to integrate gradient information into pruning criteria currently used for LLMs.
* The corresponding GBLM method is evaluated on Llama models for perplexity and zero-shot tasks.

**Strengths:**

* The paper is easy to follow and describes the proposed method in good detail.
* The method is evaluated on strong LLama models rather than older LLMs like OPT.
* Source code is provided, aiding reproducability.

**Weaknesses:**

* Integrating gradient information into pruning criteria is a well studied area, see for example [1, 2, 3, 4]. This is currently not discussed under Related Work.
* Consequently, the novelty of GBLM is quite limited. For instance, the analysis in Section 2.3 is very similar to derivations presented in [2]. Ultimately, GBLM seems to be a minor variation of a diagonal Fisher scheme (using both gradients and activations while slightly tweaking norms in a heuristic manner).
* The most robust form of evaluation, perplexity, shows only very slight improvements relative to prior work of < 0.1 points, while dropping noticably from the baseline. I am not sure if this is a significant enough improvement in practice.
* It is unclear how the gradient calculation impacts the speed and compute/memory requirements of the pruning process. Being fast and memory efficient is one of the key strengths of SparseGPT and Wanda, hence I think a detailed comparison/discussion of this aspect would be important.

Unfortunately, at this time, I find neither the method itself nor the empirical results interesting enough to recommend acceptance.

[1] Pruning convolutional neural networks for resource efficient inference, Molchanov et al.

[2] WoodFisher: Efficient Second-Order Approximation for Neural Network Compression, Singh et al.

[4] The Optimal BERT Surgeon: Scalable and Accurate Second-Order Pruning for Large Language Models, Kurtic et al.

[3] Movement Pruning: Adaptive Sparsity by Fine-Tuning, Sanh et al.

**Questions:**

* See weaknesses, in particular the compute/memory efficiency point.

---

> ### Author Response · Authors · 2023-11-23
>
> Thank you for your valuable comments and feedback, they are very helpful for us to improve our paper. We are glad that you found our paper easy to follow and detailed. In the following, we have first stated your comments and provided our response.
>
>     Q1. Integrating gradient information into pruning criteria is a well studied area, see for example [1, 2, 3, 4]. This is currently not discussed under Related Work.
>
> We thank the reviewer for bringing to our notice these important papers where gradient information is used for pruning. We have provided a discussion of these works and promise to incorporate them in the related work section of our revised paper.
>
> Although the use of gradients has been studied in the context of pruning, earlier methods [1,3] used gradients in the context of transfer learning. [1] uses gradients of the finetuning objective with respect to = feature map to calculate the importance of the feature map in CNN architecture. [3] uses gradients of the finetuning objective with respect to weight matrix for the BERT model to determine which weights that are moving away from 0 as important. Both [1,3] follow an iterative pruning finetuning schedule to obtain a pruned model that preserves the accuracy of the finetuned downstream task. This is not computationally feasible for large language models with billions of parameters.
>
> Meanwhile, [2,4] uses gradient information to estimate the Fisher matrix as an efficient approximate computation of the inverse of the Hessian for the OBS framework. Both these methods require storing gradients for inverse Fisher calculation in memory at the point of pruning, which is no feasible for large transformer architecture like the LLaMA series of models with billions of parameters. Additionally, inverse Fisher computation has a quadratic complexity on the number of parameters in the model, which makes it computationally inefficient for large language models.
>
> [1] Pruning convolutional neural networks for resource efficient inference, Molchanov et al.
>
> [2] WoodFisher: Efficient Second-Order Approximation for Neural Network Compression, Singh et al.
>
> [3] Movement Pruning: Adaptive Sparsity by Fine-Tuning, Sanh et al.
>
> [4] The Optimal BERT Surgeon: Scalable and Accurate Second-Order Pruning for Large Language Models, Kurtic et al.

---

> ### Author Response · Authors · 2023-11-23
>
> Q2. Consequently, the novelty of GBLM is quite limited. For instance, the analysis in Section 2.3 is very similar to derivations presented in [2]. Ultimately, GBLM seems to be a minor variation of a diagonal Fisher scheme (using both gradients and activations while slightly tweaking norms in a heuristic manner).
>
> The reviewer notes that the novelty of GBLM-Pruner is perceived as limited, given the existing studies [1,2,3,4] that have explored the utilization of gradient information for pruning neural networks. To address this concern, it is crucial to emphasize that our work is the first attempt to study the use of gradients for one-shot pruning of language models with billions of parameters while maintaining the zero-shot generalization capabilities of the language models to diverse downstream tasks. Unlike [1,3], where the gradients are obtained for the finetuning objective on a downstream task, our work uses the language modelling objective on a small set of calibration samples to obtain the gradients. Additionally, unlike [1,3], GBLM-Pruner does not require any weight update to maintain the model performance. This makes our proposed method computationally efficient and applicable for large language models with billions of parameters like LLaMA-1-30B and LLaMA-2-70B.
>
> Furthermore, [2,4] ignore the first-order terms under the assumption that $\Delta L ≈ 0$ and cease to provide any useful information when higher-order terms are under consideration. We illustrated how gradient information can be integrated with higher order Hessian term $(H = X^TX + \lambda I)$ or activations by using scaling term $\alpha$ to balance the magnitude difference between activations and gradients. This gives us superior performance than Wanda across different models with considerable performance gain for 2:4 sparsity pruning.
>
> To underscore the significance of the scaling term, we have incorporated an ablation analysis in the table below. It presents the perplexity values obtained for the LLaMA-2-7B model across various $\alpha$ values.
>
> | Scaling term, $\alpha$ | Perplexity |
> |----------|----------|
> | 1 |  6.912 |
> | 10 |  6.890 |
> | 100   | 6.858 |
> | 1000   | 6.902 |
> | 10000   | 6.926 |
>
> To substantiate our proposed metric, we revisited and refined the OBS framework by incorporating the consideration of gradients, i.e., the first-order term of the Taylor series expansion. The closed-form solution, derived from analyzing the increase in error when removing weight from the model, serves as the foundational basis for our novel gradient-based pruning metric. The theoretical analysis presented in [2,3] overlooks the first-order terms. Instead, it focuses on optimizing the computation of the inverse Hessian by utilizing the empirical Fisher matrix with respect to gradients.

---

> ### Author Response · Authors · 2023-11-23
>
> Q3. The most robust form of evaluation, perplexity, shows only very slight improvements relative to prior work of < 0.1 points, while dropping noticably from the baseline. I am not sure if this is a significant enough improvement in practice.
>
> The reviewer notes that the improvement achieved by GBLM-Pruner appears to be modest. To address this concern, it's essential to highlight that the enhancement demonstrated by GBLM-Pruner in unstructured pruning is comparable to Wanda's performance over SparseGPT, which is 0.08, 0.06 and 0.03 on LLaMA-2-7B/13B/70B respectively. Additionally, the benefit of the use of gradient is evident for restrictive 2:4 semi-structured pruning. Although SparseGPT tends to exhibit a superior performance, it is computationally expensive. Our proposed GBLM-Pruner outperforms Wanda by an appreciable margin while maintaining a comparable computation time, as shown in our following response. For instance, GBLM-Pruner outperforms Wanda by 0.53 perplexity score for LLaMA-2-7B and 0.20 perplexity score for  LLaMA-2-13B.
>
> To strengthen our argument regarding the benefits of 2:4 semi-structured pruning, it is essential to emphasize that 2:4 sparsity not only enhances storage efficiency but also contributes to improved inference speed. Notably, a 2:4 sparsity ratio can yield a speedup of 1.24x for the LLaMA architecture, as demonstrated by Wanda [5]. The table below illustrates the evaluation of inference speedup for 2:4 sparsity on NVIDIA A6000 GPUs across various transformer layers of the LLaMA model.
>
> | LLaMA layer | Dense | 2:4  | Speedup |
> |----------|----------|----------|----------|
> | q/k/v/o_proj | 3.49   | 2.14   | 1.63x   |
> | up/gate_proj   | 9.82   | 6.10   | 1.61x  |
> | down_proj   | 9.92   | 6.45   | 1.54x  |

---

> ### Author Response · Authors · 2023-11-23
>
> Q4. It is unclear how the gradient calculation impacts the speed and compute/memory requirements of the pruning process. Being fast and memory efficient is one of the key strengths of SparseGPT and Wanda, hence I think a detailed comparison/discussion of this aspect would be important.
>
> We thank the reviewer for pointing us to this important question. To answer this question, we would like to point out that our pruning algorithm consists of two stages. In the first stage, we compute the gradients of the pretrained language model on the language modelling objective using the calibration dataset. These precomputed gradients are then utilized in the second stage to perform layerwise pruning based on the pruning metric, $W_m[i,j] = |W[i,j]|.(\alpha \cdot ||G[:,i,j]||_p + ||X[:,j]||_2)$.
>
> The first stage entails a tradeoff between memory and speed, where increased GPU memory permits the computation of gradients with improved time efficiency through larger batch sizes. In our experiments, accounting for the GPU memory constraint across all models, we computed gradients using a batch size of 1. Also, the size of the calibration set used is very small, with only 128 samples, thereby rendering the computational cost reasonable. To illustrate the minimal compute gradient computation time, we also present the memory and compute time requirements for the 7B and 13B models on NVIDIA A100 GPUs in the table below.
>
> |Models  | Memory in GB | Time in seconds  |
> |----------|----------|----------|
> | 7B  |  62 |  71.316 |
> | 13B | 96 |  102.12 |
>
> Additionally, we want to point out that gradients only need to be computed once and can be reused. This is particularly beneficial in scenarios where iterative pruning is necessary to find the optimal sparsity for the model. Furthermore, we are uniformly pruning the language model across all layers, but if we want to study the sensitivity of each individual layer and set the pruning ratio of each layer accordingly, precomputed gradients can be efficiently reused in such cases as well.
>
> The algorithmic complexity of computing the pruning metric for GBLM-Pruner with given weight, activations, and gradient is $O(d_{hidden}^2)$. This is the same as that of Wanda. To support our argument, we have also conducted an empirical comparison between our method, SparseGPT, and Wanda in terms of computing the pruning metric and weight removal, which is depicted in the table below. The computation time is measured on NVIDIA A100 GPUs and reported in milliseconds.
>
> | Method | 7B | 13B  |
> |----------|----------|----------|
> | SparseGPT |  256732.95 | 423550.22   |
> | Wanda |  257.04 | 455.39  |
> | Ours   | 403.76 |  561.30 |
>
> The table reveals the computational efficiency of our method in comparison to SparseGPT. This efficiency is particularly notable because SparseGPT necessitates iterative weight pruning followed by weight updates, which significantly inflate its overall compute time.

---

### Official Review · Reviewer_k6s5 · 2023-11-01

**Soundness:** 2 fair
**Presentation:** 4 excellent
**Contribution:** 2 fair
**Rating:** 5
**Confidence:** 4

**Summary:**

This study introduces GBLM-Pruner, a new post-training pruning technique designed for large language models, which leverages gradient information. The authors provide both theoretical rationale and empirical assessments that demonstrate GBLM-Pruner outperforms other prominent baselines, such as Wanda and SparseGPT.

**Strengths:**

- The paper is well-organized, effectively presenting the method with clear descriptions and comprehensive empirical evaluations.
- Both theoretical explanations and empirical results are presented to validate the theoretical explanations and empirical results.
- The paper includes plenty of ablation studies, encompassing diverse sparsity levels, different pruning metrics, assessments of dependency on calibration samples, and visualizations that highlight the specifics of sparse patterns.

**Weaknesses:**

- The improvements achieved by GBLM-Pruner, as compared to other baselines like SparseGPT and Wanda, appear to be relatively modest. For instance, in Table 2, under 50% unstructured sparsity, GBLM-Pruner (l1) yields perplexity reductions of only 0.06, 0.09, and 0.05 compared to Wanda on LLaMA-2-7B/13B/70B, respectively. Additionally, in Figure 2, the curves for Wanda and GBLM-Pruner exhibit significant overlap.

- I'm unclear about the rationale behind experimenting with the pruning metrics listed in Line 7/8. It seems that some of these metrics may not provide meaningful insights.

- It's essential to understand the memory and time requirements during the pruning process of GBLM-Pruner. Obtaining gradient information can impose a significant memory cost, and it may not be feasible to conduct this process in a layer-wise manner. Storing intermediate features for the backward process could further impact memory usage. Thus, it would be valuable to compare these memory and time requirements with those of other baseline methods for a more comprehensive assessment of GBLM-Pruner's practicality.

**Questions:**

None

---

> ### Author Response · Authors · 2023-11-23
>
> Thank you for your valuable comments and feedback, they are very helpful for us to improve our paper. We are glad that you found our paper well-organized, clear and comprehensive. In the following, we have first stated your comments and provided our response.
>
>     Q1. The improvements achieved by GBLM-Pruner, as compared to other baselines like SparseGPT and Wanda, appear to be relatively modest. For instance, in Table 2, under 50% unstructured sparsity, GBLM-Pruner (l1) yields perplexity reductions of only 0.06, 0.09, and 0.05 compared to Wanda on LLaMA-2-7B/13B/70B, respectively. Additionally, in Figure 2, the curves for Wanda and GBLM-Pruner exhibit significant overlap.
>
> The reviewer notes that the improvement achieved by GBLM-Pruner appears to be relatively modest. To address this concern, it's essential to highlight that the enhancement demonstrated by GBLM-Pruner in unstructured pruning is comparable to Wanda's performance over SparseGPT, which is 0.08, 0.06 and 0.03 on LLaMA-2-7B/13B/70B respectively. Additionally, the benefit of the use of gradient is evident for restrictive 2:4 semi-structured pruning. Although SparseGPT tends to exhibit a superior performance, it is computationally expensive. Our proposed GBLM-Pruner outperforms Wanda by an appreciable margin while maintaining a comparable computation time, as shown in our following response. For instance, GBLM-Pruner outperforms Wanda by 0.53 perplexity score for LLaMA-2-7B and 0.20 perplexity score for  LLaMA-2-13B.
>
>
>     Q2. I'm unclear about the rationale behind experimenting with the pruning metrics listed in Line 7/8. It seems that some of these metrics may not provide meaningful insights.
>
> We believe line 7/8 refers to pruning metric $(|W|\cdot||X||_2)^2 - |W|\cdot|G|$  in Table 5. In conducting this experiment, we aimed to observe the impact of aggregating gradients through subtraction for the pruning metric. As shown by the very high perplexity score, it completely disrupts the pruning metric.
>
>
>     Q3. It's essential to understand the memory and time requirements during the pruning process of GBLM-Pruner. Obtaining gradient information can impose a significant memory cost, and it may not be feasible to conduct this process in a layer-wise manner. Storing intermediate features for the backward process could further impact memory usage. Thus, it would be valuable to compare these memory and time requirements with those of other baseline methods for a more comprehensive assessment of GBLM-Pruner's practicality.
>
> The reviewer suggests that using gradients in GBLM-Pruner may incur a substantial computational cost and restrict layer-wise pruning. We would like to clarify that there appears to be a misunderstanding. Our pruning algorithm does not impose a significant computational burden, and, in fact, all our pruned models undergo layer-wise pruning.
>
> To elaborate further, our pruning algorithm consists of two stages. In the first stage, we compute the gradients. These precomputed gradients are then utilized in the second stage to perform layerwise pruning based on the pruning metric, $W_m[i,j] = |W[i,j]|.(\alpha \cdot ||G[:,i,j]||_p + ||X[:,j]||_2)$.
>
> In our experiments, accounting for the GPU memory constraint across all models, we computed gradients using a batch size of 1 but can be more efficient using large batch size. To illustrate the minimal compute gradient computation time, we also present the memory and compute time requirements for the 7B and 13B models on NVIDIA A100 GPUs.
>
> |Models  | Memory in GB | Time in seconds  |
> |----------|----------|----------|
> | 7B  |  62 |  71.316 |
> | 13B | 96 |  102.12 |
>
> Additionally, we want to point out that gradients only need to be computed once and can be reused in case of iterative pruning for finding optimal sparsity. Also, the size of the calibration set used is very small, with only 128 samples, thereby rendering the computational cost reasonable.
>
> The algorithmic complexity of computing the pruning metric for GBLM-Pruner with given weight, activations, and gradient is $O(d_{hidden}^2)$, which is same as Wanda. To support our argument, we have also conducted an empirical comparison between our method, SparseGPT, and Wanda in terms of computing the pruning metric and weight removal, which is depicted in the table below. The computation time is measured on NVIDIA A100 GPUs and reported in milliseconds.
>
> | Method |7B |13B  |
> |----------|----------|----------|
> | SparseGPT |256732.95 |423550.22 |
> | Wanda |257.04 |455.39 |
> | Ours   |403.76 |561.30|
>
> The table reveals the computational efficiency of our method in comparison to SparseGPT. This efficiency is particularly notable because SparseGPT necessitates iterative weight pruning followed by weight updates, which significantly inflate its overall compute time.
>
> [1] Sun, Mingjie et al. “A Simple and Effective Pruning Approach for Large Language Models.” ArXiv abs/2306.11695 (2023): n. pag.

---

### Meta-Review · Area_Chair_82d9 · 2023-12-06

**Metareview:**

This study introduces GBLM-Pruner, a new post-training, gradient-based pruning technique designed for LLMs. The paper has a clear presentation, detailed empirical evaluations, and the inclusion of extensive ablation studies. However, the reviewer pointed out multiple concerns including limited novelty and underwhelming empirical improvements over Wanda and SparseGPT. The paper falls below the acceptance threshold due to these overarching concerns.

**Justification For Why Not Higher Score:**

- The improvements offered by GBLM-Pruner are modest, with only slight reductions in perplexity compared to existing methods. This marginal improvement questions the practical significance of the new method. Furthermore, as recently been pointed out, perplexity is an insensitive metric to compare compressed LLMs, further questioning whether the gains are meaningful in real tasks

- The novelty of the method is limited, as integrating gradient information into pruning criteria is already a well-explored area. The paper fails to adequately distinguish GBLM-Pruner from existing techniques, and reviewers consider its methodological contributions to be variations of established schemes.

**Justification For Why Not Lower Score:**

N/A

---

### Decision · Program_Chairs · 2024-01-16

Reject